# Social-BiGAT: Multimodal Trajectory Forecasting using Bicycle-GAN and Graph Attention Networks

**Vineet Kosaraju**[1*]    **Amir Sadeghian**[1,2*]    **Roberto Martín-Martín**[1]    **Ian Reid**[3]
**S. Hamid Rezatofighi**[1,3]    **Silvio Savarese**[1]

[1]Stanford University    [2] Aibee Inc    [3] University of Adelaide
vineetk@stanford.edu

## Abstract

Predicting the future trajectories of multiple interacting agents in a scene has become an increasingly important problem for many different applications ranging from control of autonomous vehicles and social robots to security and surveillance. This problem is compounded by the presence of social interactions between humans and their physical interactions with the scene. While the existing literature has explored some of these cues, they mainly ignored the multimodal nature of each human's future trajectory. In this paper, we present *Social-BiGAT*, a graph-based generative adversarial network that generates realistic, multimodal trajectory predictions by better modelling the social interactions of pedestrians in a scene. Our method is based on a graph attention network (GAT) that learns reliable feature representations that encode the social interactions between humans in the scene, and a recurrent encoder-decoder architecture that is trained adversarially to predict, based on the features, the humans' paths. We explicitly account for the multimodal nature of the prediction problem by forming a reversible transformation between each scene and its latent noise vector, as in Bicycle-GAN. We show that our framework achieves state-of-the-art performance comparing it to several baselines on existing trajectory forecasting benchmarks.

## 1    Introduction

For a variety of applications, accurate pedestrian trajectory forecasting is becoming a crucial component. Autonomous vehicles such as self-driving cars, and social robotics such as delivery vehicles must be able to understand human movement to avoid collisions [1–4]. Intelligent tracking and surveillance systems used for city planning must be able to understand how crowds will interact to better manage infrastructure [5–8]. Trajectory prediction is also becoming crucial enabling downstream tasks, such as tracking and re-identification [9]. However, trajectory prediction is still a challenging task because of several properties inherent to human behavior:

- **Social Interactions** When humans move in public, they often interact socially with other pedestrians [10]. From taking actions to avoid collisions, to walking in groups, there are several ways humans interact while moving that require prediction methods to model social behavior [11, 12]. These social interactions may not be necessarily influenced by people's spatial proximity.

- **Scene Context** Pedestrian behavior is not only dependent on the people around them, but is also highly dependent on the physical scene around them [12–16]. This includes not just stationary obstacles that cannot be avoided, such as buildings, but also different physical cues present visually, such as sidewalks or grass which may enable or restrict human movement.

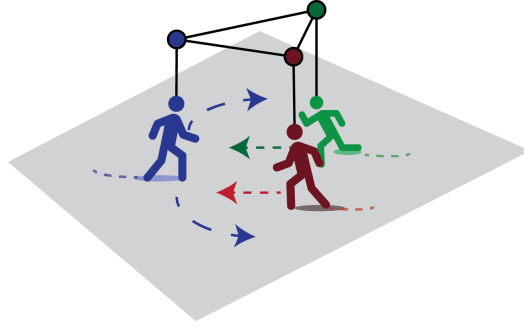

Figure 1: We show multimodal behavior for the blue pedestrian, who must make a decision about which direction they will take to avoid the red-green pedestrian group.

- **Multimodal Behavior** Pedestrians may follow several plausible trajectories, as there is a rich distribution of potential human behavior [10, 11, 17, 18]. For example, when two pedestrians are walking towards each other, several modes of behavior develop, such as moving to the left or moving to the right. Within each mode, there is also a large variance, allowing pedestrians to vary features like their speed.

Prior work in trajectory forecasting have tackled several of the previously listed challenges and have informed our architectural design. Helbing *et al.* [19] and Pellegrini*et al.* [20] successfully demonstrated the benefit of modeling social interactions but require handcrafted rules that are less able to generalize to new scenes. Alahi *et al.* [10] utilized recurrent architectures to consider multiple timesteps of pedestrian behavior, but do not consider the physical cues of the scene. Other prior research has also focused on understanding the physical scene. Lee *et al.* [15] and Sadeghian *et al.* [16] use raw scene images and soft attention on the scene to highlight important cues. Their work is limited by not considering social cues jointly with the scene.

By contrast, Gupta *et al.* [11] and Sadeghian *et al.* [12] utilize GANs with social mechanisms that do take into account all people in the scene. However both models fall short of learning the truly multimodal distribution of human behavior, and instead learn a single mode of behavior with high variance. Further, both models are limited by how they learn social behavior: while the former loses information by using the same social vector for all pedestrians in a scene, the latter requires a hand-defined sorting operation that may not perform optimally in all cases.

To address the limitations of these works, we propose Social-BiGAT, a GAN [21] based approach to construct a generative model that can learn these essential multimodal trajectory distributions. The main contributions of this work are as follows. First, we improve the modeling of social interactions between pedestrians in a scene by introducing a flexible graph attention network [22], where all pedestrians in a scene are allowed to interact. This improves over prior works where either interactions were limited locally, or interactions were modelled using hand-defined rules. Next, we encourage generalization towards a multimodal distribution by constructing a reversible mapping between outputted trajectories and latents that represent the pedestrian behavior in a scene, as previously performed for images by Zhu *et al.* [23]. This allows us to generate trajectories that are socially and physically acceptable, while also learning a larger multimodal trajectory distribution, despite only having access to single samples from single modes of behavior across scenes. Finally, we incorporate physical scene cues using soft attention as in [12, 16] to make our model more generalizable.

## 2 Related Work

In recent years due to the rise of popularity in development of autonomous driving systems and social robots, the problem of trajectory forecasting has received significant attention from many researchers in the community. The majority of existing works have been focused on the effects of incorporating physical features of the scene into human-space models [15, 16], as well as learning how to model social behavior between pedestrians in human-human models [10, 24]. Other works have approached the problem from a generative setting [11] and have jointly modeled these features in one framework [12]. While these works have greatly advanced the field, they have drawbacks that we address by incorporating graph attention networks [22] and image translation networks [23].

**Trajectory Forecasting** Traditionally, pedestrian trajectory prediction has been tackled by defining handcrafted rules and energy parameters that capture human motion but fail to generalize properly [19, 20, 24–26]. Instead of handcrafting these features, modern approaches rely on recurrent neural networks that learn these parameters directly from the data [10, 16], while incorporating some means of capturing human interaction features [15, 27, 28]. Several of these prior methods have been limited in scope, as they often limit interactions to nearby pedestrian neighbors [10, 29, 30] and do not model global interactions or cannot generalize to a variable number of humans. Other approaches have explored trajectory prediction from a generative standpoint, including Lee *et al.* [15], Gupta *et al.* [11], and Sadeghian *et al.* [12], with their own limitations. The former only considers interactions within a limited local scope, and the latter two result in models with high variances. Specifically, although human motion is inherently multimodal, these methods are not able to expressively learn this multimodal behavior and instead learn one mode with a high variance. In our work we incorporate ideas from image to image translation to generate multimodal pedestrian trajectories. Furthermore, our model uses graph attention networks [22] to more efficiently and robustly model the interactions between the agents in the scene, whereas prior research [12, 31] depend on hand-defined rules.

**Graph Attention Networks** Proposed by Velickovi *et al.* [22], graph attention networks (GAT) allow for the application of a self-attention based architecture over any type of structured data that can be represented as a graph. These networks build upon the prior advances of graph convolutional networks (GCN) [32] by also allowing for the model to implicitly assign different importances to nodes in the graph. In our case, we can formulate pedestrian interactions as a graph, where nodes refer to human humans, and edges are these interactions; higher edge weights correspond to more important interactions. By leaving the graph fully connected, we can model interactions between humans without using pooling [11] or sorting [12] that may lose important features.

**Image Translation** The field of image domain translation has gone through several seminal advancements in the past couple years. The first advancement was made with the *pix2pix* framework [33], which enabled translation but was limited by requiring paired training examples. Zhu *et al.* improved this model with CycleGAN [34], which was able to learn these domain mappings with unpaired examples from each domain through a cycle consistency loss. Newer research has focused on learning multimodality of the output: InfoGAN [35] focuses on maximizing variational mutual information, while BicycleGAN [23] introduces a latent noise encoder and learns a bijection between noise and output. In our model we draw upon the advancements suggested by BicycleGAN to propose a latent space encoder that allows for multimodal pedestrian trajectory generation.

## 3 Social-BiGAT

### 3.1 Problem Definition

Formally defined, human trajectory prediction is the problem of predicting the future navigation movements of pedestrians (namely their $x$ and $y$ coordinates on a 2D map representation), given their prior movements and additional contextual information about the scene. We assume the route taken by each pedestrian is influenced by the location of other humans and the physical constraints on its path, as well as its own goal, which is to some extent encoded in its past course of movements. For any particular scene, the inputs to our model are twofold: 1) scene information, in the form of a top-down or side-view image of the scene, $I^t$, and 2) the previously observed trajectory within the scene of each of the $N$ currently visible pedestrians, $X_i = \left\{ (x_i^t, y_i^t) \in \mathbb{R}^2 | t = 1, \ldots, t_{obs} \right\}$ for $\forall i \in \{1, \ldots, N\}$.

Given all above inputs and the ground truth future trajectory for each pedestrian between $t_{pred}$ and $t_{obs}$ timesteps, *i.e.* $Y_i = \left\{ (x_i^t, y_i^t) \in \mathbb{R}^2 | t = t_{obs} + 1, \ldots, t_{pred} \right\}$ for $\forall i \in \{1, \ldots, N\}$, our goal is to learn the underlying (and potentially, multimodal) distribution which can generate feasible samples for their future trajectories, *i.e.* $\hat{Y}_i$ for $\forall i \in \{1, \ldots, N\}$.

### 3.2 Overall Model

Our overall model consists of four main networks, each of which is made up of three key modules (Figure 2). Specifically, we construct a generator, two forms of discriminators (one that operates at local pedestrian scale, and one that operates at a global scene-level scale), and a latent space encoder. Our generator is composed of a feature encoder module (Section 3.3), an attention network module (Section 3.4), and a decoder module (Section 3.5). The feature encoder module extracts encodings from raw features for use in the attention network, which in turn learns which features are most

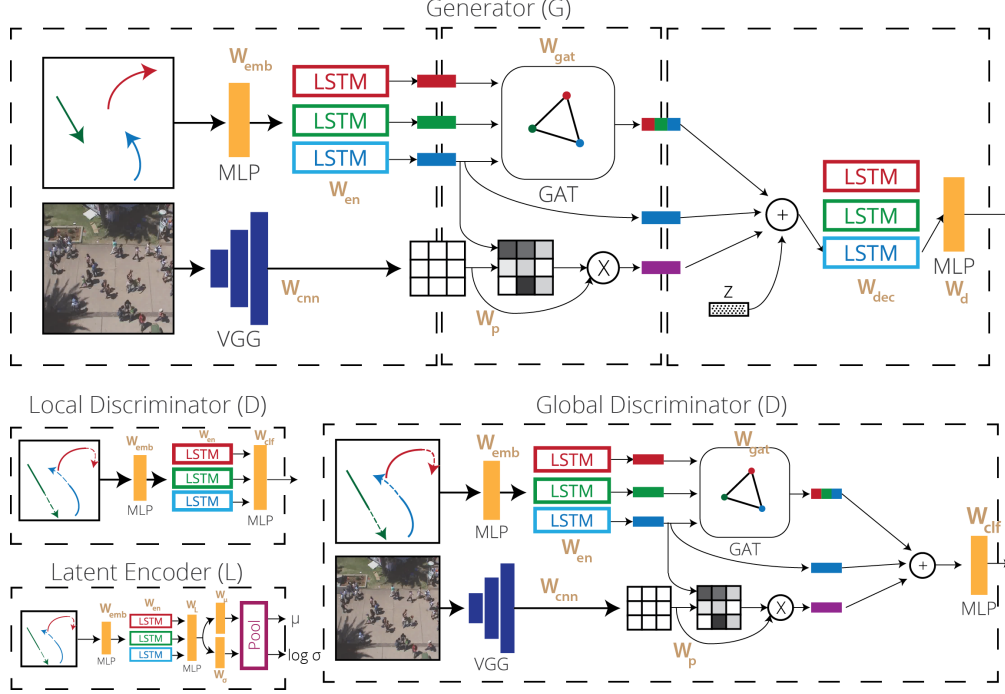

Figure 2: Architecture for the proposed Social-BiGAT model. The model consists of a single generator, two discriminators (one at local pedestrian scale, and one at global scene scale), and a latent encoder that learns noise from scenes. The model makes use of a graph attention network (GAT) and self-attention on an image to consider the social and physical features of a scene.

important in generation. These weighted features are then passed into the decoder module, which uses LSTMs to generate multiple timesteps of trajectories. The architecture is trained adversarially with both discriminators, as motivated by Isola *et al.* [33] and to encourage realistic local and global trajectories, and we also train a latent scene encoder that learns to generate a mean and variance for the noise that best represents a scene jointly, as in Zhu *et al.* [23] to encourage multimodality.

## 3.3 Feature Encoder

The feature encoder has two main components: a social pedestrian encoder, in order to learn representations of observed pedestrian trajectories, and a physical scene encoder, in order to learn the representation of the scene features. For the social encoder, for each pedestrian we first embed the pedestrian's relative displacements into a higher dimension using a multilayer perceptron (MLP), and then encode these pedestrian movements across timesteps into a single embedding using a LSTM, resulting in encoding $V_s(i)$ for pedestrian $i$. For the physical feature encoder, we simply pass the top-down image view of the scene through a convolutional neural network (CNN), resulting in $V_p$:

$$V_s(i) = LSTM_{en}(MLP_{emb}(X_i, W_{emb}), h_{en}(i); W_{en}) \tag{1}$$

$$V_p = CNN(I; W_{cnn}) \tag{2}$$

## 3.4 Attention Network

Much like how humans intuitively know which other pedestrians to focus on to avoid collisions, we want our model to better understand the relative weight that interactions have: we accomplish this goal by applying attention over our extracted features.

**Physical Attention** To apply attention over our physical features relative to a specific pedestrian, we take in $V_s(i)$, and apply soft attention, where the network is parameterized by $W_p$ and outputs context vector $C_p{}^t(i)$:

$$C_p(i) = ATT_p(V_p, V_s(i); W_p) \tag{3}$$

**Social Attention** Similar to physical attention, we use as input to our social attention model the embeddings of pedestrians, $V_s(i)$. The social attention model encodes pedestrians as weighted (attended) sum of the neighbor pedestrians they interact with. Prior research has used either permutation invariant symmetric functions, such as *max* or *average* [11], or ordering functions such as sorting based on euclidean distance [12]. In the former, the downside is that each pedestrian receives an identical joint feature representation that discards some uniqueness. While the latter technique does not suffer from this drawback, it does require setting a maximum number of pedestrians and does impose a human bias on the model that is not necessarily always true. Namely, it assumes that euclidean distance ordering is a key component of understanding social interactions.

To avoid these flaws, we utilize graph attention networks [22, 36]. Given pedestrian $i$'s embedding, $V_s(i)$, for all pedestrians in the scene, we apply several stacked graph attention layers. Each layer, $\ell$, is applied as follows, where $W_{gat}$ parameterizes a shared linear transformation and $a$ is the shared attentional mechanism:

$$e_{ij} = a(W_{gat}V_s(i), W_{gat}V_s(j)) \tag{4}$$

$$\alpha_{ij} = \text{softmax}_j(e_{ij}) \tag{5}$$

$$C_s{}^\ell(i) = \sum_{j \in N} \alpha_{ij} W_{gat} V_s(j) \tag{6}$$

We use the features $C_s^L$ from the last GAT layer where $\ell = L$ as the final social features. We allow the graph of pedestrians to remain fully connected and do not apply any mask. This allows each pedestrian to interact with each other and does not impose any restriction on pedestrian orders.

## 3.5 GAN Network

In this section we present how our feature encoder and attention network serve as core building blocks in developing the LSTM based Generative Adversarial Network (GAN). GANs typically consist of two networks that compete with each other: a generator, and a discriminator. While the generator learns to generate realistic samples from input data, the discriminator learns to discern which samples are real, and which are generated, thereby engaging in a two-player min-max game.

**Generator** The generator is built using a decoder LSTM. Similar to conditional GANs [37], our generator takes as input a noise vector $z$ sampled from a multivariate normal distribution, and is conditioned on the physical scene context, $C_p(i)$, the pedestrian scene context, $C_s{}^L(i)$, and the previous pedestrian encoding, $V_s(i)$. These are all concatenated together such that $C_g(i) = [V_s(i), C_s{}^L(i), C_p(i), z]$. Generation of trajectories across multiple timesteps is then performed through a decoder LSTM, such that:

$$\hat{Y}_i = MLP_d(LSTM_{dec}(C_g(i), h_{dec}(i); W_{dec}); W_d) \tag{7}$$

**Discriminator** The discriminator architecture mirrors that of the generator, with encoder LSTMs used to represent pedestrians, and a CNN used to represent scene features. We propose two versions of this core discriminator architecture: one at local scale, operating on pedestrians, and one at global scale, operating on an entire scene. The former performs classification directly on encodings of concatenated past and future trajectories, such that:

$$\hat{L}(i) = MLP_{clf}(LSTM_{en}(MLP_{emb}([X_i, \tilde{Y}_i], W_{emb}), h_{en}(i); W_{en}); W_{clf}), \tag{8}$$

where $\tilde{Y}_i \sim p(Y_i, \hat{Y}_i)$ is a randomly chosen future trajectory sample from the either ground truth or predicted path. $\hat{L}(\cdot)$ is classification score representing the sample is a ground truth (real) or predicted (fake) with the truth label $L(i) = 1$ and $L(i) = 0$, respectively.

The global discriminator performs the same classification operation, but on the global context vector for the pedestrian trajectory; namely, the concatenation of the physical scene context, $C_p(i)$, the pedestrian scene context, $C_s{}^L(i)$, and the pedestrian encoding, $V_s(i)$.

## 3.6 Latent Encoder

In order to generate trajectories that are truly multimodal, we encourage our model to develop a bijection between the outputted trajectories and the latent space inputted to the generator. Specifically,

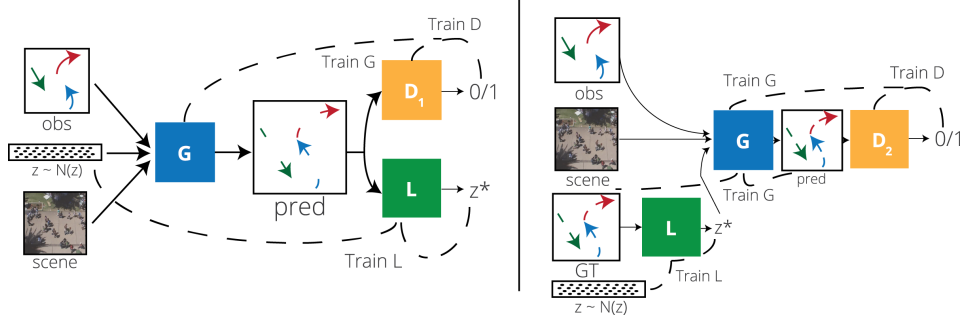

Figure 3: Training process for the Social-BiGAT model. We teach the generator and discriminators using traditional adversarial learning techniques, with an additional L2 loss on generated samples to encourage consistency. We further train the latent encoder by ensuring it can recreate noise passed into the generator, and by making sure it mirrors a normal distribution.

we want to map both the latent noise to an output trajectory, as well as map that trajectory back to the original latent. While the former task is accomplished by a generator, we perform the latter using a latent scene encoder, as previously performed in Zhu *et al.* [23].

The architecture for the latent scene encoder is relatively similar to the local discriminator. First, pedestrians are encoded in the scene using a LSTM encoder. Embeddings from this LSTM are passed in two parallel MLPs that are trained to output a mean $\mu_i$ and log variance $\sigma^2{}_i$ for each pedestrian:

$$\mu_i = MLP_\mu(MLP_L(V_s(i), W_L), W_\mu) \tag{9}$$

$$\log \sigma^2{}_i = MLP_\sigma(MLP_L(V_s(i), W_L), W_\sigma) \tag{10}$$

Means and log variances across pedestrians are max pooled together to generate a single mean and log variance representation of the latent for a given scene.

## 3.7 Losses

As illustrated in Figure 3, to train these four models we have a multistep training process, where we not only perform a transformation starting from the noise, $z \to \hat{Y}_i \to \hat{z}$, but also perform a transformation starting from the trajectories, $Y_i \to z \to \hat{Y}_i$. In the former we have two main loss terms to consider: the GAN loss ($L_{gan_1}$) from the generator fooling the discriminator, and the discriminator correctly classifying the generator, as well as a loss term on reconstructing the noise ($L_z$). We calculate these as follows, where $G$ refers to the generator, $D$ to the discriminator and $E$ to the latent encoder:

$$L_{gan_1} = \mathbb{E} \log D(X_i, Y_i) + \mathbb{E} \log(1 - D(X_i, \hat{Y}_i)) \tag{11}$$

$$L_z = ||E(\hat{Y}_i) - z||_1 \tag{12}$$

In the latter, we have three additional loss terms: the GAN loss ($L_{gan_2}$), a L2 loss on trajectories ($L_{traj}$), enforcing the generation of real samples, and a KL loss on the generated noise ($L_{kl}$) such that it resembles noise drawn from a random Gaussian:

$$L_{gan_2} = \mathbb{E} \log D(X_i, Y_i) + \mathbb{E} \log(1 - D(X_i, G(X_i, E(Y_i)))) \tag{13}$$

$$L_{traj} = ||Y_i - G(X_i, E(Y_i))||_2 \tag{14}$$

$$L_{kl} = \mathbb{E}[D_{kl}(E(Y_i)||N(0, I))] \tag{15}$$

We ultimately combine all these loss terms using $\lambda$ weights that are chosen as hyperparameters:

$$G*, D*, E* = \underset{G,E}{\text{argmin}} \, \underset{D}{\text{argmax}} [L_{gan_1} + \lambda_z L_z + L_{gan_2} + \lambda_{traj} L_{traj} + \lambda_{kl} L_{kl}] \tag{16}$$

## 4 Experiments

We perform experiments on two relevant datasets: ETH [20] and UCY [38]. Both contain annotated trajectories of socially interacting pedestrians in real world scenes. The datasets include different types

| Dataset | Discriminative | | Generative | | Ours | | |
|---|---|---|---|---|---|---|---|
| | Lin | S-LSTM | S-GAN-P | Sophie | GAT | BiGAN | Social-BiGAT |
| ETH | 1.33 / 2.94 | 1.09 / 2.35 | 0.87 / 1.62 | 0.70 / 1.43 | **0.68 / 1.29** | 0.72 / 1.47 | **0.69 / 1.29** |
| HOTEL | **0.39 / 0.72** | 0.79 / 1.76 | 0.67 / 1.37 | 0.76 / 1.67 | 0.68 / 1.40 | 0.54 / 1.12 | 0.49 / 1.01 |
| UNIV | 0.82 / 1.59 | 0.67 / 1.40 | 0.76 / 1.52 | **0.54 / 1.24** | 0.57 / 1.29 | **0.55 / 1.34** | **0.55 / 1.32** |
| ZARA1 | 0.62 / 1.21 | 0.47 / 1.00 | 0.35 / 0.68 | **0.30 / 0.63** | **0.29 / 0.60** | 0.32 / 0.65 | **0.30 / 0.62** |
| ZARA2 | 0.77 / 1.48 | 0.56 / 1.17 | 0.42 / 0.84 | 0.38 / 0.78 | **0.37 / 0.75** | 0.49 / 0.88 | **0.36 / 0.75** |
| AVG | 0.79 / 1.59 | 0.72 / 1.54 | 0.61 / 1.21 | 0.54 / 1.15 | 0.52 / 1.07 | 0.52 / 1.09 | **0.48 / 1.00** |

Table 1: Baseline models compared to our architectures when predicting 12 future timesteps, given the previous 8. Errors reported are ADE / FDE in meters, with generative models being evaluated using $K = 20$ samples.

| Model | K = 20 | K = 10 | K = 5 | K = 1 | % Increase |
|---|---|---|---|---|---|
| S-GAN-P | 0.558 / 1.118 | 0.594 / 1.214 | 0.650 / 1.316 | 0.846 / 1.758 | 51.6% / 57.2% |
| Sophie | 0.526 / 1.030 | 0.566 / 1.122 | 0.604 / 1.266 | 0.712 / 1.456 | 35.3% / 41.4% |
| GAT | 0.518 / 1.064 | 0.529 / 1.127 | 0.584 / 1.241 | 0.682 / 1.494 | 31.6% / 40.4% |
| BiGAN | 0.523 / 1.091 | 0.531 / 1.144 | 0.579 / 1.298 | 0.662 / 1.439 | **26.6% / 31.9%** |
| Social-BiGAT | **0.476 / 0.998** | **0.488 / 1.096** | **0.527 / 1.260** | **0.606 / 1.328** | 27.3% / 33.1% |

Table 2: Effect of varying $K$ in evaluation results for generative models. We see that reducing $K$ results in a higher average ADE/FDE across the five scenes for S-GAN-P and Sophie, due to higher distribution variances.

of social interactions, ranging from group formation to collision avoidance, the type of interaction we aim to encode with our Social-BiGAT model. The datasets contain five unique scenes: Zara1, Zara2, Univ, Eth, and Hotel. We evaluate Social-BiGAT on these datasets and compare to several deterministic baselines, including a linear regressor that minimizes least square error, *Linear*, and a predictive model using LSTMs and social pooling, *S-LSTM* [10], as well as two main generative models: *S-GAN-P*, which applies generative modeling to social LSTMs [11], and *Sophie*, which applies attention networks to social GANs [12]. We present evaluation results of three versions of our model: one trained without the latent scene encoder but with the graph attention network, *GAT*, one trained without the graph attention network but with the latent scene encoder, *BiGAN*, and our final model with all components included, *Social-BiGAT*. Models are evaluated using two main metrics: average displacement error (ADE), and final displacement error (FDE). Both are defined as the average L2 distance between the ground truth and predicted trajectories. Evaluation occurs over a timescale of 8 seconds, where the first 3.2 seconds (8 timesteps) correspond to observed data, and the last 4.2 seconds (12 timesteps) correspond to predicted future data. We evaluate using a hold-one-out cross evaluation strategy in meter space, with $N$-$K$ variety loss, as previously performed [11, 12].

## 4.1 Quantitative Results

We compare our model to various baselines in Table 1, reporting the average displacement error (ADE) and final displacement error (FDE) for 12 timesteps of pedestrian movement. As expected we see that both the discriminative Social LSTM baseline outperforms the simple linear model, and that the generative baselines, which are evaluated from $K = 20$ samples, improve upon the discriminative ones by generating a full distribution of possible human trajectories. In terms of our proposed architectures, we see that incorporating the GAT alone does indeed improve performance, as the network is able to more flexibly account for pedestrian interactions. Alternatively the BiGAN alone does not help performance. Our combined GAT and BiGAN architecture, Social-BiGAT, does however achieve the best performance of all our models, resulting in a $0.15$ meter decrease in average FDE from the previous state-of-the-art model. This is due to the reduced errors for the *Hotel* scene compoared to other generative architectures.

While the BiGAN architecture does not help performance much when $K = 20$, we show in Table 2 that it does help improve generalization at lower settings of $K$. Specifically, while S-GAN-P and Sophie suffer from higher variances, causing their ADE and FDE to increase dramatically when $K$ is lowered, Social-BiGAT's ADE and FDE increase more slowly. Further, we see that the GAT architecture initially performs better than the BiGAN, but with fewer samples the GAT error increases faster. This aligns with our intuition that the inclusion of the BiGAN in our architecture enables for better capturing of a multimodal distribution, instead of generating samples from a unimodal distribution. This suggests that the inclusion of the latent scene encoder in BiGAN and Social-BiGAT allow for the architecture to reduce the variance of the outputted trajectory distributions while also allowing for better generalization.

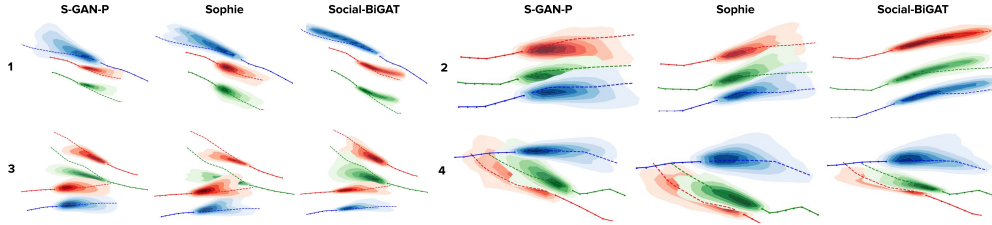

Figure 4: Generated trajectories visualized for the S-GAN-P, Sophie, and Social-BiGAT models across four main scenes. Observed trajectories are shown as solid lines, ground truth future movements are shown as dashed lines, and generated samples are shown as contour maps. Different colors correspond to different pedestrians.

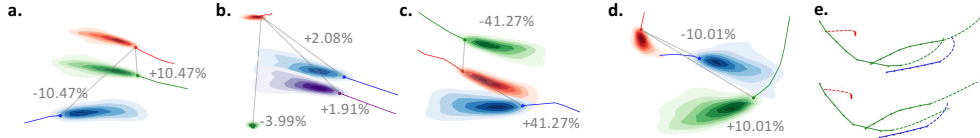

Figure 5: Visualizations of Social-BiGAT behavior, with attention weights shown with respect to the red agent in (a, b, c, d) and latent space exploration in scene (e). The attention weights derive social features beyond distance, such as collision avoidance, while adjusting the latents modifies the aggressiveness and speed of agents.

## 4.2 Qualitative Results

In order to better understand the contribution of the graph attention network and bicycle training structure in improving understanding of social behavior, we visualize the generated trajectories for four scenes, comparing our proposed Social-BiGAT model to S-GAN-P and Sophie (Figure 4). We draw three main conclusions from these visualizations. First, as shown in scenes 1 and 2, Social-BiGAT often has a lower variance than S-GAN-P and Sophie, suggesting that it can generate more efficiently. Second, as shown in scenes 2 and 3, the model is better able to model the interactions of people travelling in crowds or groups. Finally, as scene 4 demonstrates, the model can generate realistic trajectories for pedestrians that are attempting to avoid collisions. Each of these findings are crucial in ensuring that the model performs optimally across a wide range of social behavior.

In addition to visualizing our model's trajectories in comprison to prior generative baselines, we also depict the attention weights of our model and the impact of modifying $z$ while keeping the scene fixed in Figure 5. In scenes (a, b), the attention weight roughly lines up with Euclidean distance. Scenes (c, d) show that the attention further generalizes in learning which agents are important socially: in Scene (c) it pays large attention to the blue agent it may collide with in the future, even though that agent is farther away from it than the green one, and in Scene (d) it ignores the blue agent for the farther green agent with whom it might collide with. Finally in (e), we adjust the latent $z$ resulting in behavior between the blue and green agents that ranges from cautious (top) to aggressive (bottom).

## 5 Conclusion

We presented Social-BiGAT, a novel architecture for forecasting pedestrian movements that outperforms prior state-of-the-art methods across several widely used trajectory benchmarks. Unlike prior research, our model is able to generate multiple trajectories for multiple humans in a multimodal fashion. Through our evaluations and visualizations we demonstrated that Social-BiGAT is able to capture the intricate social nature of pedestrian movements and that we are able to control the predictions by adjusting the latents at test time. We further introduced several important architectural improvements to the generation process: 1) we utilize a social attention graph network (GAT) to better learn pedestrian interactions through the data, and 2) we train using two discriminators that operate at local and global scale. As shown experimentally, with these design patterns our Social-BiGAT model is able to generate pedestrian trajectories that more realistically predict human motion.

## 6 Acknowledgement

The research reported in this publication was supported by funding from the TRI gift, ONR (1165419-10-TDAUZ), Nvidia, and Samsung.

## Footnotes

*indicates equal contribution

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
