[Supplementary Material · submission_v3_supplementary.pdf]

# Social-BiGAT: Supplementary Material

## 1 Implementation Details

**Feature Encoder** For the physical feature vector's CNN network, we used a VGG19 model [1] that has been fine-tuned for scene segmentation [2]. This method has been applied with prior success in several prior works [3, 4].

**Attention Network** As suggested in Vaswani *et al.* [5], we also stack each layer multiple times using multi-headed attention for stability, and use all of these heads as input to our next layer. We use 2 attention heads, and 2 intermediate stacked GAT layers, plus an additional final GAT output layer, for a total 3 stacked GAT layers. Prior empirical studies have demonstrated that the strength of GAT comes from multiple stacked layers [6]. We use a setting of $\alpha = 0.2$ and when performing a forward pass through these multiple stacked layers, we take the average of the heads and apply the Elu activation function as the input to the next stacked layer. At the very last layer, we concatenate all the heads and attend over those.

**Generator** Our generator (LSTM) uses an embedding dimension of 64, and an MLP dimension of 128. Encoded trajectories are represented with a hidden dimension of 32, and decoded trajectories are represented with a hidden dimension of 32. The generator's learning rate is $1.0 \cdot 10^{-4}$ and we perform one step per training iteration.

**Discriminators** Our discriminators (LSTM) use an embedding dimension of 64, an MLP dimension of 128, and encode trajectories with hidden dimensions of 48. They are trained with a learning rate of $1.0 \cdot 10^{-3}$ and we perform one step per training iteration. Our global discriminator additionally uses a bottleneck MLP of the initial encodings that encodes trajectories to a dimension of 512 and then 8, with Relu activation functions.

**Latent Encoder** Our latent encoder and generator use a noise dimension of 8. This noise is drawn from a standard normal, such that $z \, N(0, I)$. The encoder further uses a MLP dimension of 128, and takes trajectories of dimension 32, and encodes them using this MLP into a dimension of 24. Means and variances are computed for each pedestrian, and then average pooled across pedestrians. Our network outputs the mean and log variance, for numerical stability reasons. The latent encoder is trained with a learning rate of $1.0 \cdot 10^{-3}$.

**Losses & Training** We use the Adam optimizer to train the individual G, D1, D2, and L architectures. We additionally use a linear learning rate decay schedule after the first 100 epochs, and train for 200 epochs. We iteratively train networks in isolation by first updating G and L, and then updating D1 and D2. Updating G and L consists of first updating G and L jointly, and then updating just G. Updating G and L requires computing the discriminator GAN loss from both D1 and D2 to update G, and then applying the reconstruction loss and KL loss for the latent noise to update L. We then update just G using the reconstruction loss on trajectories. Finally, we update D1 and D2 by computing the discriminator GAN loss on real and fake data and comparing to the expected scores of 0 (fake) and 1 (real). Although LSGAN loss has been suggested to train the generators and discriminators [6], we found better results when using binary cross entropy loss. When adding loss terms, we set $\lambda_{KL} = 0.05$, $\lambda_{traj} = 10.0$, and $\lambda_z = 1.0$.

Figure 1: Generated trajectories visualized for the S-GAN-P, Sophie, and Social-BiGAT models across four main scenes. Observed trajectories are shown as solid lines, ground truth future movements are shown as dashed lines, and generated samples are shown as contour maps. Different colors correspond to different pedestrians.

## 2  Note on Previous Work Comparisons

There exists a large number of works on pedestrian trajectory forecasting. We chose a select number of recent works for comparison based on code availability. Some works, such as [7] were not included in our comparison due to issues such as private source code. Other works have been trained on additional datasets and are not suitable for comparison.

## 3  Qualitative Visualizations

We present additional visualizations of generated trajectories, comparing both our proposed model to prior baselines Sophie [4] and S-GAN-P [8] (Figure 1), as well as exploring the latent dimension of our model (Figure 2). As noted in the main text of our paper, Social-BiGAT is primarily able to outperform Sophie and S-GAN-P due to a lower variance, and better representation of different modes of pedestrian behavior.

Figure 2: Visualization of generated trajectories (dashed lines), given observed trajectories (solid lines) for various (color-coded) pedestrians, while varying $z$, the noise passed into the generator. We note several modes of behavior, including avoidance versus aggressiveness (a), linearity versus curvature (b), and fast versus slow (c).