[Reviews · NeurIPS 2019]

Reviewer 1



Summary: The paper proposed to improve the interaction modeling between pedestrians by using a graph attention network [22] for the trajectory prediction task and learn multimodal trajectory distributions by using Bicycle-GAN [23]. The experimental results showed the effectiveness of the proposed approach by achieving state-of-the-art performance on the public benchmarks. Also, they showed that the performance of the proposed approach is more robust to varying K than that of the baselines, indicating that the proposed approach was successful in addressing the high variance issue in the existing approaches to a certain extent. Strengths: -- The paper is clearly written so it was easy to follow. The reasoning behind the choice of [22] and [23] for the trajectory prediction task is also clearly presented in the paper. -- The proposed approach achieved good final performance and also provided ablation study results to validate each of the proposed components. The authors' claims regarding why [22] and [23] would be effective in overcoming the limitations of the existing approaches are backed up by these results. -- The visual examples also back up the authors' claims and provide more insights about the proposed model to the readers. Weaknesses: -- Although I value the authors' observation regarding why [22] and [23] could potentially be effective for the trajectory prediction task, I feel that the novelty of this paper is limited in a sense that the authors mainly adopt and combine two existing frameworks for their task. -- The claim "While the BiGAN architecture does not help performance much when K = 20, we show in Table 2 that it does help improve generalization at lower settings of K" in lines 243-244 was not supported by Table 2. Table 2 only compares Social-BiGAT against two other baselines. In order to validate the above statement, Table 2 should also include the results from GAT and BiGAN as Table 1 did. -- I have a few questions for clarification 1) In lines 154-156, why is it the case that max or average pooling leads to each pedestrian receiving "an identical joint feature representation that discards some uniqueness". Wouldn't each pedestrian still receive a unique features that encode interactions between the selected pedestrian and other pedestrians? 2) Was Sophie the state-of-the-art approach at the time of submission? I am asking this to make sure that the proposed method was compared against strong baselines. Overall comments: I think that the technical contributions of this paper are on the NeurIPS borderline due to the fact that the authors mainly adopted the existing frameworks for a new task. However, I believe that applying these to a new problem is not trivial. The authors well justified their choice of [22] and [23] for the trajectory prediction task in the paper. Also, the authors provided enough experimental results to validate their claims and show the effectiveness of the proposed method. Thus, I believe that the paper is in good shape and includes enough new findings that an ML audience like NeurIPS would appreciate. Final comment: The authors addressed the major concerns raised by the reviewers successfully. Thus, I will keep my original score (7).

Reviewer 2



The paper aims to simulate video trajectories of multiple pedestrians in the presence of interactions. In this regard a graph based GAN method is proposed. It also integrates recent ideas such as cyclic GANs and a recurrent decoder-encoder module. Empirically it is shown to achieve promising results. It is overall a potentially interesting paper while its present form has a number of issues to be discussed below. *Problem. Instead of taking image sequences as input, the goal seems to be simplified to considering a top-view only scenario, and taking partial temporal sequence of multi-pedestrian 2D locations so far as inputs. It is better to state this clearly in the very begining to avoid potential misunderstanding or disappointment. *Presentation. **The presented algorithmic techniques are somewhat incremental. Meanwhile, the developed model, as illustrated schematically in Fig.1, is rather complicated. And it may not be intuitive why certain modules/parts are there, and thorough evaluations including w vs. w/o parts are useful to understand what is going on, and why all the parts are necessary. **Notations. As a result of a complicated model, many internal parameters are mentioned during the development, but quite a few of them lack clear and concrete definations, interpretations, and suggestion of practical values. For example, what are the definitions of the random variables W_emb, W_en in Eq.1, and W_cnn in Eq.2, and W_p in Eq.3, and W_gat in Eq.4, and W_dec & W_d in Eq.7, and W_clf in Eq.8, and all the new parameters introduced in the remaining equations 9-16. It is always expected that when a new parameter is introduced, it needs to be properly defined, its role explained, and its practical value setting set/suggested. *empirical experiments is somewhat limited. It would be much more convincing to visually present in demo videos how things are working in real image sequences, which is precisely the goal of the paper. The numbers presented in Tables 1&2 are helpful but lacking intuitions. The Figs. 4&5 are not very easy to be understood. For example, it is unclear from Fig.4 how the multi-modality nature of the multi-pedestrian motion prediction problem could be well addressed. Fig.5 seems to be an attempt in this direction, while after multiple attempts, I am still not sure whether I properly interpret the content. I will suggest the authors to at least consider providing video demos in the supplementary.

Reviewer 3



In this manuscript, the authors study the highly relevant problem of pedestrian trajectory prediction. Identifying several shortcomings of prior work in the area, they propose Social-BiGAT, which combines generative adversarial network approaches with a graph attention network (GAT) core, demonstrating strong results on a variety of established benchmarks, along with a visualisation of the interpretability in the model's learnt latent space of trajectories. I think the motivation is strong, the architecture is very sensible, and the results are strong, both from a quantitative and qualitative standpoint. I would vote for acceptance. Two suggestions for improving would concern the clarity and interpretability of the writeup: - Generally, I find the paper very well-written, and likely to invite interest from graph neural network practitioners into this application domain (which I think would be very important for further improvements). The idea of modelling the interactions between pedestrians in a scene explicitly using a GAT model is very clear and easy to motivate. ** However, I find that the architectural description of the model might be somewhat lackluster -- as the equations are not always clear to someone inexperienced in GANs or GATs, and the figures might involve too many things going on. This might make it harder for the paper's results to be appropriately reproduced or improved on. I would, in the very least, invite the authors to clearly specify in Figure 2 which arrows correspond to which variables in the equations. - As the attentional mechanism learns interaction coefficients between pedestrians, it could be highly interesting to include a qualitative study on these coefficients -- i.e. what kinds of features about pedestrians influence the attention's focus the most. I would assume that (Euclidean) distance would probably play some part, but it would be relevant to see if any additional insights can be gained from this. --- After rebuttal --- I thank the authors for their careful consideration of my review and the provided qualitative study of the attentional coefficients. This provides even further substantiation that the GAT coefficients are a useful addition to the model. With confidence that the authors will update their architectural description's clarity accordingly (keeping in mind the limited space for the rebuttal text), I am increasing my score to 9.

[Author Response · NeurIPS 2019]

**R1, R2: Technical novelty.** While our work builds on top of previously proposed methods including GAN, attention mechanisms, and GAT, their adaptation, integration and application to the complex problem of human trajectory prediction is a non-trivial technical challenge. Our main technical contribution is a novel neural model that 1) better encodes social cues (a crucial factor to predict human trajectories) based on neural graph structures, and 2) enables multimodal predictions (an intrinsic property of future human motion) inspired by BicycleGAN architecture. Learning an optimal representation of social behavior is a non-trivial task, as humans follow unwritten social rules and juggle a variety of implicit factors. As we discuss in our literature review and experimental evaluation, prior works that model social interactions fall short by either limiting the expressiveness of the models (eg: by using a pooling mechanism that is unable to capture interactions in large scenes), or by imposing human-defined constraints rather than learning from the data. Further, applying a multi-modal distribution to represent future human trajectories is not easy since it requires solving a delicate trade-off between increasing variance of each model vs. increasing multimodality, which relates to the degradation of the model's predictions as a function of allowed samples.

**R1, R3: Analysis on the attention weights.** As the reviewers suggested, we have run several experiments exploring the correlation between the weights and different pedestrian features. The figure on the right depicts the attention weights of other pedestrians wrt. the red pedestrian for different scenes. Weight values correspond to the percent decrease/increase compared to without attention: positive indicates more attention was paid to that interaction, and negative indicates less attention was paid. From Scenes (a, b) we can infer that Euclidean distance is one feature the network implicitly uses to assign attention. Scenes (c) and (d) show however that the attention further generalizes in learning which agents are important socially: in Scene (c) it pays large attention to the blue agent it may collide with in the future, even though that agent is farther away from it than the green one, and in Scene (d) it ignores

the blue agent for the farther green agent with whom it might collide with. The diversity of social awareness that the model displays validates the choice of GAT over prior works. We will include this figure and a visualization of physical attention to the final manuscript, with more samples in the supplementary material.

**R1: Improved generalization claim.** We thank R1 for their suggestion and have updated the table as shown below. We see that the GAT architecture initially performs better than the BiGAN, but with fewer samples the GAT error increases faster. This aligns with our intuition that the inclusion of the BiGAN in our architecture enables for better capturing of a multimodal distribution, instead of generating samples from a unimodal distribution. We note that BiGAN's error increases slightly slower than Social-BiGAT, but the inclusion of the GAT allows for a lower overall error.

**R1: Comparison with SOTA baselines.** Yes, Sophie was the state of the art in terms of ADE/FDE at the time of submission.

**R1: Clarifying pooling for joint feature representation.**

| Model | K=20 | K=10 | K=5 | K=1 | % Increase |
|---|---|---|---|---|---|
| GAT | 0.518 / 1.064 | 0.529 / 1.127 | 0.584 / 1.241 | 0.682 / 1.494 | 31.6% / 40.4% |
| BiGAN | 0.523 / 1.091 | 0.531 / 1.144 | 0.579 / 1.298 | 0.662 / 1.439 | **26.6% / 31.9%** |
| Social-BiGAT | **0.476 / 0.998** | **0.488 / 1.096** | **0.527 / 1.260** | **0.606 / 1.328** | **27.3% / 33.1%** |

In architectures that use pooling, each pedestrian is encoded into a single vector. Across all pedestrians the vectors are pooled, resulting in a single vector representing the social interactions between all agents in the scene (not specific to any particular agent), which each pedestrian receives. Our choice of GAT serves as a major improvement as it allows for different interactions to be attended to based on their social importance, which the network learns.

**R2: Inputting top-view image instead of image sequences.** Thanks to the reviewer for pointing out the confusion, we will clarify this in the paper. However, we would like to point out that most prior research/datasets in this field are using a single top-down image input, such as CAR-Net, Desire, and Sophie, and the more recent Social Ways, and Multi Agent Tensor Flow. Moreover, our approach has also been tested on sequences where the scene's angle view is not necessarily perfectly top-view, e.g. UCY dataset. The criticism of not accepting a sequences of images is valid, as we only use a single image. This is done for fair comparison, as prior methods also only use single images. Finally, our work is indeed limited by requiring temporal sequences of pedestrian locations. However, while we agree that performing end-to-end tracking and forecasting directly on sequences of images is a very good future direction , we are currently limited by a lack of proper benchmark data. All existing benchmarks follow the same input modality, and we do the same to make the comparison feasible. In the future we definitely hope to research this same problem end-to-end.

**R2, R3: Lack of clarity in notations, architecture.** We will clarify the equations and better explain the variables we introduce before using them, along with labelling weights in Fig. 2, which should improve the architecture explanation. We also will add an in-depth ablative analysis and explanation of model components in the supplementary material.

**R2: Better visualizations & demos.** As suggested by R2, we have already started implementing it by adding several examples to the supplementary work, where we can visualize multiple dimensions of the latent space across multiple pages and better illustrate how our model qualitatively results in multimodality. We will also add figures exploring how the latent noise generated through the encoder changes when varying parts of the scene (such as number of agents, or speed and direction of agents). We also are looking forward to publishing demo videos that we can place in the supplementary material as well. Space permitting, we will add these figures to the main paper as well.

[Meta-Review · NeurIPS 2019]

The authors have presented a strong rebuttal and improved the clarity of the manuscript with the responses to the suggestions of the reviewers. This paper opens up a new avenue for graph neural network researchers, with a clearly motivated architecture which achieves solid results, both quantitatively and, following the rebuttal response, also qualitatively. In discussion with the reviewers regarding the paper and the rebuttal, the main concern remained the lack of a video demo, which would greatly improve the quality of this manuscript. However, considering the strong performance numbers presented in the paper, it would be surprising if the qualitative result in the demo video turns out to be terrible or does not make any sense. Also, the authors included convincing qualitative examples in the paper and the rebuttal already, which kind of addressed the concerns of the major reviewers. We also agreed that notational issues can be fixed through minor editing.